# Compositional Modeling to Analyze the Effect of CH$_4$ on Coupled Carbon Storage and Enhanced Oil Recovery Process

**Jinhyung Cho [1], Gayoung Park [2], Seoyoon Kwon [2], Kun Sang Lee [3], Hye Seung Lee [3] and Baehyun Min [2,*]**

[1]  Severe Storm Research Center, Ewha Womans University, 52 Ewhayeodae-gil, Seodaemun-gu, Seoul 03760, Korea; jh.cho@ewha.ac.kr

[2]  Department of Climate and Energy Systems Engineering, Ewha Womans University, 52 Ewhayeodae-gil, Seodaemun-gu, Seoul 03760, Korea; gypark7049@ewhain.net (G.P.); sykwon024@ewhain.net (S.K.)

[3]  Department of Earth Resources and Environmental Engineering, Hanyang University, 222 Wangsimni-ro, Seongdong-gu, Seoul 04763, Korea; kunslee@hanyang.ac.kr (K.S.L.); seung7185@hanyang.ac.kr (H.S.L.)

*  Correspondence: bhmin01@ewha.ac.kr; Tel.: +82-2-3277-6946

**Abstract:** The present study is aimed at the development of compositional simulation models of the co-injection of CO$_2$ and CH$_4$ during the water-alternating-gas (WAG) process in order to assess the efficiency of carbon capture and storage in combination with enhanced oil recovery (CCS-EOR). The co-injection of CO$_2$ and CH$_4$ occupies more reservoir pore volume and causes higher reservoir pressure than CO$_2$ WAG, thus leading to an enhanced early EOR performance. However, the overall EOR performance of the co-injection method becomes lower than that of CO$_2$ WAG due to the reduced miscibility and sweep efficiency upon further CH$_4$ addition. The decrease in gas displacement and sweep efficiency weaken the hysteresis effects upon the residual trapping mechanism. However, the solubility trapping mechanism takes effect because the co-injection generates higher average reservoir pressure than does the CO$_2$ WAG. The index of global warming potential (GWP) in a mole unit is employed to quantify the carbon storage effects of CO$_2$ and co-injection WAG cases. According to the index, 1 mole of CH$_4$ sequestration has the same effects as that of 10 moles of CO$_2$ for global warming mitigation. In conclusion, the carbon storage effects are enhanced as CH$_4$ concentration in the WAG increases.

**Keywords:** carbon capture and storage associated with enhanced oil recovery (CCS-EOR); CH$_4$; water alternating gas (WAG); global warming potential (GWP)

## 1. Introduction

Carbon capture and storage (CCS) has become well-known as a technology for reducing the emission of greenhouse gases from fossil fuels during power generation and industrial processes [1]. Projects involving the injection of carbon dioxide (CO$_2$) for enhanced oil recovery (EOR) have been operating worldwide since the early 1970s [2] and EOR is expected to be a major driver for CCS by providing an additional revenue stream. Hence, combining the CO$_2$-EOR and CCS into a coupled CCS-EOR approach provides a synergistic effect towards business and environmental protection by offering commercial opportunities to oilfield operators. Consequently, almost 80 million tons (Mt) of CO$_2$ are already being used for CCS-EOR each year [3,4].

Methane (CH$_4$) is generated as a by-product of oil extraction during primary and secondary oil production and EOR. As CH$_4$ is a potent greenhouse gas (GHG), the operator often chooses to flare this by-product instead of releasing it directly to the atmosphere if there is little commercial opportunity or

sufficient regulatory incentive to bring it to market [5]. Also, $CH_4$ has been used as a re-injection gas for EOR [6,7]. Previous studies have examined the impacts of impurities in the $CO_2$ stream upon the minimum miscibility pressure (MMP) to find that the presence of $CH_4$ increases the MMP compared to that of $CO_2$, thus negatively impacting upon EOR performance [8–14]. Moreover, since the global warming potential (GWP) of $CH_4$ over 100 years is 28 times that of $CO_2$ [15], it is worth considering the use of $CH_4$ for CCS projects aimed at mitigating global warming and climate change. Although some studies on the geological storage of $CO_2$ and $CH_4$ in an aquifer have been conducted, there are insufficient studies examining the impact of $CH_4$ on CCS-EOR [16–19]. Hence, it is necessary to store $CH_4$ in reservoirs in order to mitigate gas flaring and release into the atmosphere by using a CCS-EOR approach that can consider energy security and climate change simultaneously. However, $CH_4$ affects the miscibility as well as the hysteresis and solubility effects for residual and solubility trapping mechanisms and the ultimate EOR performance during CCS-EOR.

For this reason, compositional simulation models are developed in the present study in order to investigate the effects of $CO_2$-$CH_4$ co-injection upon the CCS-EOR mechanisms and performance. The multiple-mixing-cell method is applied to calculate the MMP of the injected gases (i.e., a mixture of $CO_2$ and $CH_4$) and the reservoir oil. Of the four main $CO_2$ trapping mechanisms (namely: structural, residual, solubility, and mineral trapping), structural and mineral trapping are excluded from the scope of the present study due to their negligible effects upon CCS-EOR in a relatively short period [20]. Here, $CO_2$ can be stored in the reservoir by hysteresis and dissolution in water during the water-alternating-gas (WAG) process, thus indicating residual and solubility trapping mechanisms. A three-phase hysteresis model and Henry's law are applied for the residual trapping and solubility trapping, respectively.

## 2. Methodology

### 2.1. Calculation of Minimum Miscibility Pressure (MMP)

Carbon dioxide ($CO_2$) is the most common type of gas used in gas injection and has been widely used in the EOR method for light oils. Injected $CO_2$ acts as a solvent to reduce oil viscosity and expand oil volume in reservoirs. These phenomena occur more frequently under miscible conditions than they do under immiscible conditions [21]. As shown schematically in Figure 1, miscibility conditions during $CO_2$-EOR can be achieved via multiple contact processes that make use of condensing and vaporizing gas drives. At the injected $CO_2$ front, intermediate molecular weight hydrocarbons evaporate from the reservoir oil into the $CO_2$, where they equilibrate with the $CO_2$ to take part in the next contact with the reservoir oil in a phenomenon called the vaporizing gas drive or the forward multiple contact process. After each contact, however, the equilibrated reservoir oil behind the injected $CO_2$ close to the injector is continuously mixed with fresh $CO_2$ in a process called the condensing gas drive or the backward multiple contact process [22,23]. Due to the mass transfer of intermediate hydrocarbons between $CO_2$ and oil, the composition and properties, such as the density and viscosity, of the reservoir oil and injected gases are changed. This, in turn, leads to a difference in the interfacial tension (IFT) between the equilibrated oil and gas at each location. As the reservoir approaches the miscible condition, the oil viscosity and IFT decrease, such that the IFT tends towards zero and the miscible condition is reached with a reservoir pressure higher than the MMP [24].

A variety of methods, such as the slim-tube, vanishing interfacial tension, and rising bubble technique, have been conventionally employed to estimate the MMP [25–27]. However, since these methods are time-consuming, expensive, and occasionally less accurate, it is often beneficial to implement computational methods for calculating the MMP. For this purpose, the multiple-mixing-cell method using the cubic equation of state (EOS) is one of the most prevalent methods [28]. In the present study, this method is applied according to the following steps:

1. The system temperature is defined and the starting pressure for the process is estimated.
2. A calculation is performed for the displacing gas and the displaced oil to obtain the new equilibrium compositions of liquid and vapor after the first contact.

3.  Step 2 is repeated for each contact using the previous and new equilibrium compositions, injected gas, and reservoir oil to obtain the updated compositions until the lengths of all key tie-lines converge to a tolerance of $10^{-8}$.
4.  The tie-line length is computed for the pressure and the minimum tie-line length (TL) is saved.
5.  The pressure is increased and Steps 2–4 are repeated.
6.  A multiple-parameter TL regression is performed to determine the exponent $n$, the slope $m$, and the constant $b$ in the equation $TL^n = mp + b$ (power-law extrapolation), and the resulting function is plotted.
7.  The MMP is determined when the power-law extrapolation gives zero to within the desired accuracy of 20 psia at the latest three pressures.

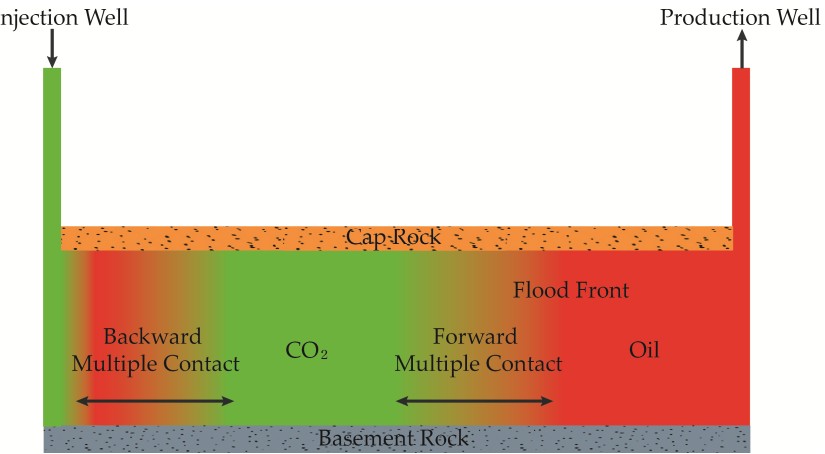

**Figure 1.** Schematic diagram of the multiple contact miscibility process during $CO_2$-enhanced oil recovery (EOR).

## 2.2. The Three-Phase Hysteresis Model for Residual Trapping

Although conventional two-phase hysteresis models assume the reversibility of permeability during the drainage process following the previous imbibition cycle, this assumption is invalid for three-phase fluid flow because the relative permeability of gas is lower when displacing a water-oil mixture than when displacing oil only [29]. The three-phase hysteresis model proposed by Larsen and Skauge [30] is adopted in the present study and has shown better matching to experimental data than the two-phase model [31,32].

The gas permeability during the drainage process is calculated using Equation (1):

$$k_{rg}^{drain}\left(S_g,\ S_w^I, S_g^{start}\right) = \left[k_{rg}^{input}\left(S_g\right) - k_{rg}^{input}\left(S_g^{start}\right)\right]\left(\frac{S_{wi}}{S_w^I}\right)^\alpha + k_{rg}^{imb}\left(S_g^{start}\right) \tag{1}$$

We clarify that a list of symbols is given in the Nomenclature.

The relative permeability of the gas with decreasing gas saturation (i.e., non-wetting phase in the imbibition process) is denoted by $S_{gf}$ and is estimated by Equations (2) and (3):

$$k_{rg}^{imb}\left(S_g\right) = k_{rg}^{drain}\left(S_{gf} + S_g^{end}\right) \tag{2}$$

$$S_{gf} = \frac{1}{2}\left\{\left(S_g - S_{gr}\right) + \sqrt{\left(S_g - S_{gr}\right)^2 + \frac{4}{C}\left(S_g - S_{gr}\right)}\right\} \tag{3}$$

where the trapped gas saturation, $S_{gr}$, and Land's parameter [33], $C$, are estimated from Equations (4) and (5), respectively:

$$S_{gr} = S_{gc} + \frac{S_{gm} - S_{gc}}{1 + C\big(S_{gm} - S_{gc}\big)} \tag{4}$$

$$C = \frac{1}{S_{gr,\mathrm{max}} - S_{gc}} - \frac{1}{S_{g,\mathrm{max}} - S_{gc}} \tag{5}$$

$S_{gr,\mathrm{max}}$ is the maximum trapped gas saturation and can be approximated using an empirical correlation with the porosity, $\phi$ [34], in accordance with Equation (6):

$$S_{gr,\mathrm{max}} = 0.5473 - 0.9696\phi \tag{6}$$

To determine the relative permeability of the oil, the Stone's first model modified by Aziz and Settari [35] is applied. The minimum residual oil saturation, designated $S_{om}$, is modified to reflect the effect of trapped gas on oil relative permeability, as shown in Equation (7):

$$S_{om}^{\mathrm{mod}} = S_{om} - a\big(S_g - S_{gf}\big) \tag{7}$$

where $S_g$ and $S_{gf}$ are the total and free gas saturations respectively, and $a$ is input parameter of 0.68 for mimicking a strongly water-wet condition [36].

### 2.3. Solubility Model

Carbon dioxide ($CO_2$) is also trapped by the solubility trapping mechanism due to contact with water. The mole fraction of $CO_2$ in each phase is determined from the thermodynamic equilibrium condition according to Equation (8):

$$f_{i,o} = f_{i,g} = f_{i,w} \text{ with } i = 1, \ldots, n_c \tag{8}$$

where $f_{i,o}$, $f_{i,g}$, and $f_{i,w}$ denote the fugacity of the $i$-th component in the oil, gas, and water phase, respectively.

$f_{i,o}$ and $f_{i,g}$ are calculated using Peng-Robinson EOS [37,38]. By contrast, $f_{i,w}$ is given by Henry's law [39] in accordance with Equation (9):

$$f_{i,w} = y_{i,w} H_i \text{ with } i = 1, \ldots, n_c \tag{9}$$

where $H_i$ and $y_{i,w}$ are Henry's constant and the mole fraction of the $i$-th component in the water phase, respectively. Henry's constant is calculated using Equations (10) and (11) under isothermal conditions [40,41]:

$$\ln H_i = \ln H_i^S + \frac{1}{RT} \int_{p_{H_2O}^s}^{p} \overline{v_i}\, dp \tag{10}$$

where $R$ is the gas constant and $T$ is the temperature in Kelvin. Then,

$$\ln H_i^s = \ln p_{H_2O}^s - D\big(T_{r,H_2O}\big)^{-1} + E\big(1 - T_{r,H_2O}\big)^{0.355}\big(T_{r,H_2O}\big)^{-1} + F\exp\big(1 - T_{r,H_2O}\big)\big(T_{r,H_2O}\big)^{-0.41} \tag{11}$$

where $D$, $E$, and $F$ are constants with values of $-9.4234$, $4.0087$, and $10.3199$, respectively, for $CO_2$, and $\overline{v_i}$ is the partial molar volume of the $i$-th component at infinite dilution ($cm^3 \cdot mol^{-1}$), which is computed using Equation (12) [42]:

$$\overline{v}_{CO_2} = -47.75418 + 4.336154 \times 10^{-1} T - 5.945771 \times 10^{-4} T^2 \tag{12}$$

where $T$ is the reservoir temperature in K.

## 3. Results

### 3.1. Fluid Modeling

Weyburn W3 fluid data [43,44] is used as the experimental data for fluid modeling in the present study. Since the Weyburn Field in Saskatchewan, Canada, reached the economic limit of waterflooding and became a target for $CO_2$-EOR, extensive experiments have been conducted to investigate the interactions between oil and $CO_2$ for $CO_2$-EOR and CCS purposes [43,45].

The fluid composition of W3 experimental data and corresponding input EOS parameters used for fluid modeling are summarized in Table 1, while Table 2 presents the oil properties calculated via a regression method using the Peng-Robinson Equation of State (PR-EOS) and matched against the experimental data to increase the reliability of the compositional reservoir simulation [37,38]. The MMP is then estimated using the multiple-mixing-cell method [28]. An examination of the bottom row of Table 2 indicates that the MMP datum associated with $CO_2$ in W3 fluid at a reservoir temperature of 63 °C is 14,196 kPa (second column, Table 2), while the calculated value is 13,872 kPa (third column, Table 2), thus indicating a 2.3% error against the W3 data.

**Table 1.** Fluid composition of W3 experimental data and properties of each component used for the EOS calculation.

| Component | Composition | Critical Pressure $p_c$ (kPa) | Critical Temperature $T_c$ (K) | Acentric Factor | Molecular Weight |
|---|---|---|---|---|---|
| $N_2$ | 0.0207 | 3394.4 | 126.2 | 0.040 | 28.0 |
| $CO_2$ | 0.0074 | 7376.5 | 304.2 | 0.225 | 44.0 |
| $H_2S$ | 0.0012 | 8936.9 | 373.2 | 0.100 | 34.1 |
| Methane, $C_1$ | 0.0749 | 4600.2 | 190.6 | 0.008 | 16.0 |
| Ethane, $C_2$ | 0.0422 | 4883.9 | 305.4 | 0.098 | 30.1 |
| Propane, $C_3$ | 0.0785 | 4245.5 | 369.8 | 0.152 | 44.1 |
| Butane, $C_4$ | 0.0655 | 3722.7 | 416.5 | 0.185 | 58.1 |
| Pentane, $C_5$ | 0.0459 | 3379.4 | 464.9 | 0.239 | 72.1 |
| $C_{6-9}$ | 0.2156 | 3019.6 | 556.3 | 0.331 | 102.5 |
| $C_{10-17}$ | 0.2202 | 2017.5 | 692.2 | 0.584 | 184.0 |
| $C_{18-27}$ | 0.1027 | 1327.0 | 808.4 | 0.893 | 306.2 |
| $C_{28+}$ | 0.1252 | 1155.1 | 915.5 | 1.100 | 565.6 |

**Table 2.** Comparison of the W3 experimental data and fluid model properties.

| Parameters | W3 | Fluid Model | Difference (%) |
|---|---|---|---|
| Saturation pressure (kPa) | 4920 | 4780 | 2.85 |
| Oil density at saturation pressure (kg/m³) | 806.4 | 806.8 | −0.05 |
| Viscosity at saturation pressure (mPa·s) | 1.76 | 1.75 | 0.57 |
| Formation volume factor (m³/m³) | 1.12 | 1.11 | 0.89 |
| API (°) | 31 | 34.8 | −12.26 |
| MMP with $CO_2$ (kPa) | 14,196 | 13,872 | 2.28 |

The MMPs calculated using the multiple-mixing-cell method with varying mole fraction ratios of $CO_2$ to $CH_4$ are presented in Table 3. Here, it can be seen that the MMP increases with increasing mole fraction of $CH_4$ in the $CO_2$ injection stream, thus indicating that $CH_4$ reduces the displacement efficiency compared to that of pure $CO_2$ injection, in agreement with previous experimental studies [13,14].

**Table 3.** Minimum miscibility pressures (MMPs) calculated using the multiple mixing cell method with various mole fraction ratios of $CO_2$ to $CH_4$.

| Components | MMP (kPa) |
|---|---|
| $CO_2$ 100% | 13,872 |
| $CO_2$ 90% + $CH_4$ 10% | 21,346 |
| $CO_2$ 80% + $CH_4$ 20% | 22,015 |
| $CO_2$ 70% + $CH_4$ 30% | 22,622 |

### 3.2. Effects of $CH_4$ Injection on EOR Efficiency

A two-dimensional (2D) homogeneous model based on the work of Cho et al. [19] is designed to focus on investigating the miscibility and sweep efficiency without gravity override during WAG simulations. This reservoir model is discretized into $33 \times 33 \times 1$ grid blocks of volume $3 \times 3 \times 3$ m$^3$ each. A quarter of a 10-acre five-spot well pattern (i.e., $CO_2$ injector and oil producer) is set up for the reservoir model. Thus, the EOR and CCS performances are mainly governed by the displacement and sweep efficiencies. The total simulation period is 13 years (2007–2020), beginning with 3 years of waterflooding followed by 10 years of WAG, as shown schematically in Figure 2. The WAG consists of five cycles, in each of which water and gas are injected sequentially. Four WAG model simulation cases are analyzed, namely: 100% $CO_2$ + 0% $CH_4$ (Case 1), 90% $CO_2$ + 10% $CH_4$ (Case 2), 80% $CO_2$ + 20% $CH_4$ (Case 3), and 70% $CO_2$ + 30% $CH_4$ (Case 4). The gas injection rate for each case is a constant 2265 m$^3$/day under surface conditions to determine the effects of $CO_2$ and $CH_4$ compressibility upon EOR performance while excluding the effect of any difference in injection rate. The initial conditions for the reservoir model are presented in Table 4. Thus, the producing bottom hole pressure is fixed at 13,789 kPa, while the initial reservoir pressure and temperature are computed on the basis of hydrostatic and geothermal gradients.

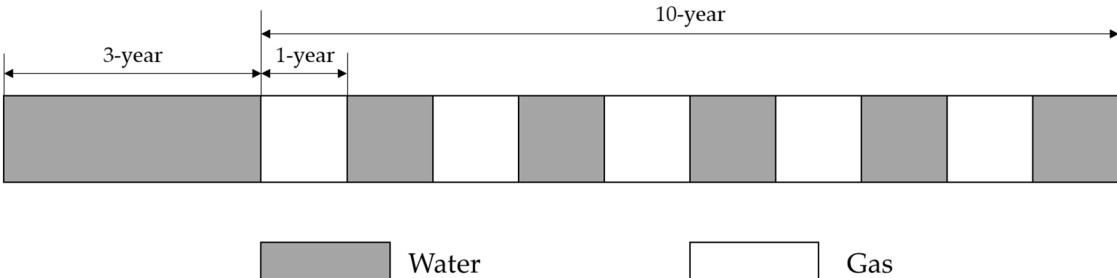

**Figure 2.** Schematic diagram of the water alternating gas (WAG) injection model.

**Table 4.** Initial and operating conditions used for the reservoir simulation.

| Parameters | Values |
|---|---|
| Depth (m) | 2811 |
| Initial reservoir pressure (kPa) | 27,579 |
| Reservoir temperature (°C) | 63 |
| Permeability (m$^2$) | $1.0 \times 10^{-13}$ |
| Porosity (%) | 0.25 |
| Initial oil saturation, $S_o$ (fraction) | 0.7 |
| Connate water saturation, $S_w$ (fraction) | 0.3 |
| Producing bottom hole pressure (kPa) | 13,789 |

The decrease in displacement efficiency due to $CO_2$-$CH_4$ co-injection is described in terms of the IFT between the displacing and displaced fluids in the middle of the reservoir (grid block co-ordinates (17, 17, 1)) during the 13-year WAG in Figure 3. Due to the lower molecular weight of $CH_4$ compared to $CO_2$, the addition of $CH_4$ makes the displacing fluid lighter and the IFT of the gas mixture higher compared to that of Case 1, as indicated in Figure 3a. In addition, the IFT after the multiple contact with the initial oil (2010–2011) is examined more closely in Figure 3b. Here, Case 1 gives the lowest IFT value of 0.08 mN/m between the $CO_2$ and reservoir oil, while Cases 2, 3, and 4 give 225%, 588%, and 1000% higher values of 0.26, 0.55, and 0.88 mN/m, respectively.

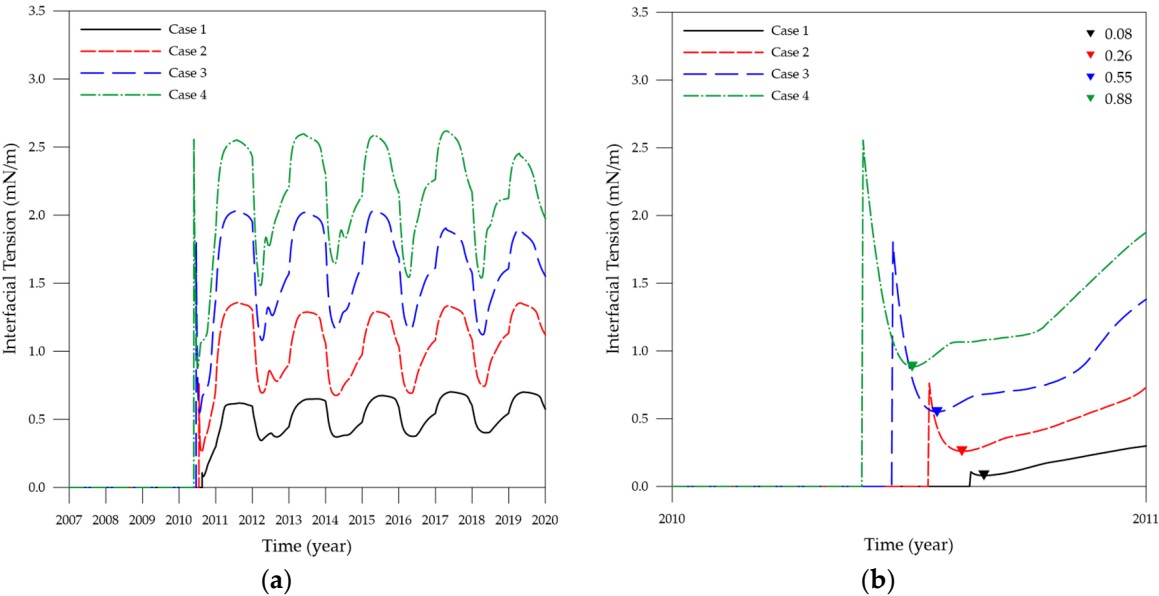

**Figure 3.** Interfacial tension (IFT) with $CH_4$ addition into the $CO_2$ stream during the WAG process under reservoir conditions (T = 63 °C) for (**a**) 2007–2020, (**b**) 2010–2011.

Meanwhile, the decrease in displacement efficiency due to $CO_2$-$CH_4$ co-injection is described in terms of the 2D change in oil viscosity when 0.15 PV of gas is injected in Figure 4. The blue swept area in Figure 4a indicates that the injected fluids make maximum contact with the reservoir oil in Case 1, and that the minimum oil viscosity in this Case is 0.46 mPa·s. When 20% mole fraction of $CH_4$ is added into the $CO_2$ injection stream (Case 3), the deep blue region of Figure 4a changes to the light blue area of Figure 4b, indicating a 23.9% increase in the oil viscosity from 0.46 mPa·s to 0.57 mPa·s. The injected gas displaces the light and intermediate oil components from the initial oil, thereby increasing the oil viscosity after the multiple contact. For Case 1, the oil viscosity near the gas injector is zero because the oil is displaced by the high injection pressure and is miscible with the $CO_2$. In addition, Figure 4 shows that the injection gas front in Case 3 is more convex than that in Case 1. This is because less gas is acting as a solvent to reduce the oil viscosity, while the fraction of injected gas that does not come into contact with the oil creates a gas channel that leads to early breakthrough of produced gas, as indicated in Figure 5.

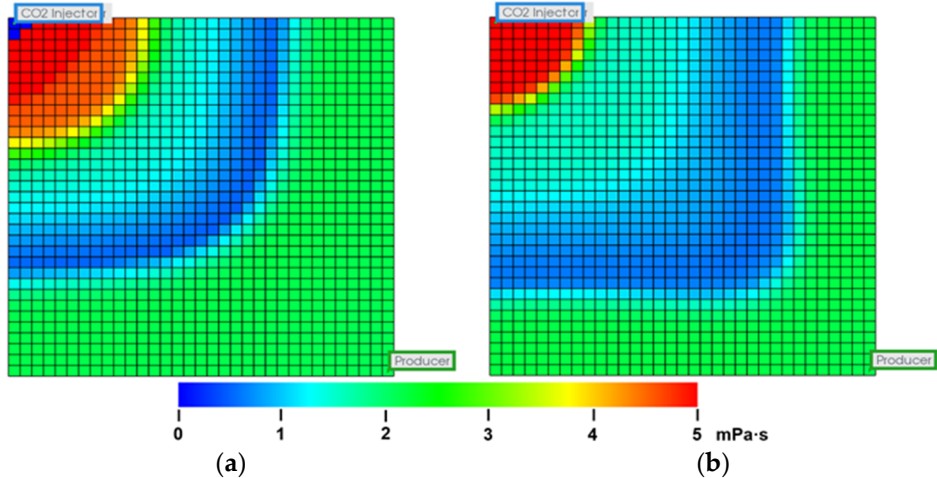

(**a**)                                        (**b**)

**Figure 4.** WAG models of oil viscosity at 0.15 PV injection for (**a**) Case 1, (**b**) Case 3.

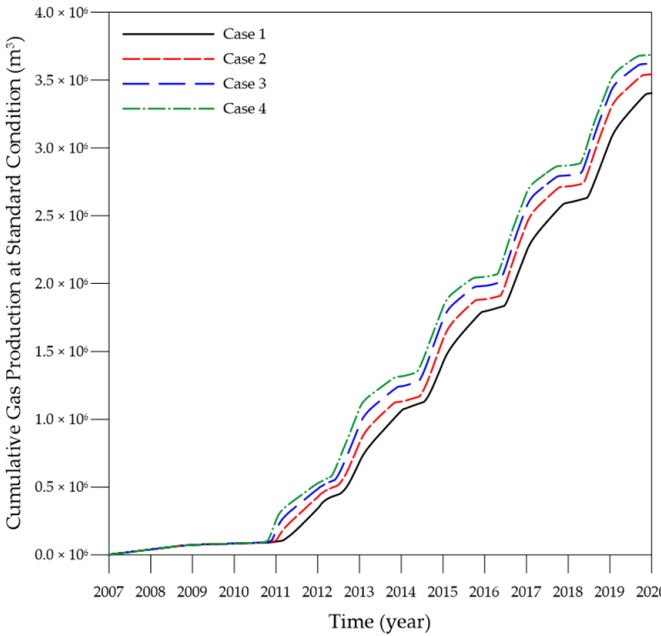

**Figure 5.** Cumulative gas production obtained from the four $CO_2$-$CH_4$ WAG cases under standard conditions.

The cumulative gas injection under the reservoir condition obtained from the four $CO_2$-$CH_4$ WAG cases is indicated in Figure 6. Note that although the gas injection rate at the surface is constant, the volume occupied by gas differs in each case because $CH_4$ is less compressible than $CO_2$; hence, the more $CH_4$ is added, the higher the gas injection rate at the reservoir (Jin et al., 2017). In addition, the average gas saturation in each of the four cases for the ten-year WAG period is indicated in Figure 7. In the first WAG cycle (2010–2012), Case 1 displays the lowest average gas saturation, while Cases 2 to 4 indicate that the average gas saturation increases with increasing addition of $CH_4$ to the gas stream as $CH_4$ occupies more reservoir pore volume than does $CO_2$. Interestingly, this trend is reversed in the subsequent WAG cycles due to the poor sweep efficiency resulting from the $CH_4$ addition, so that the average gas saturation eventually becomes the lowest for Case 4 by the end of the final WAG cycle (2019). Meanwhile, the average reservoir pressure obtained from the four $CO_2$-$CH_4$ WAG cases is indicated in Figure 8, where the greater pore occupation by $CH_4$ is seen to increase the average reservoir pressure due to the lower compressibility of $CH_4$ relative to $CO_2$. As a result, the co-injection of $CO_2$-$CH_4$ generates higher oil recovery than the pure $CO_2$ WAG during the early stages of WAG

(2010–2012), but reduces the overall oil recovery at the end of the WAG process, as indicated in Figure 9. This is because the effects of displacement and reduction in sweep efficiency become more dominant than that of compressibility as the WAG proceeds.

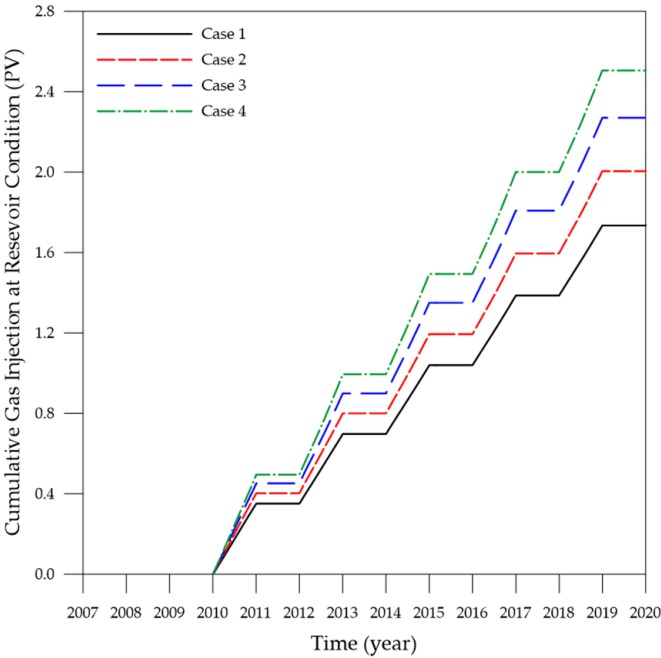

**Figure 6.** Cumulative gas injection obtained from the four $CO_2$-$CH_4$ WAG cases at the reservoir condition (T = 63 °C).

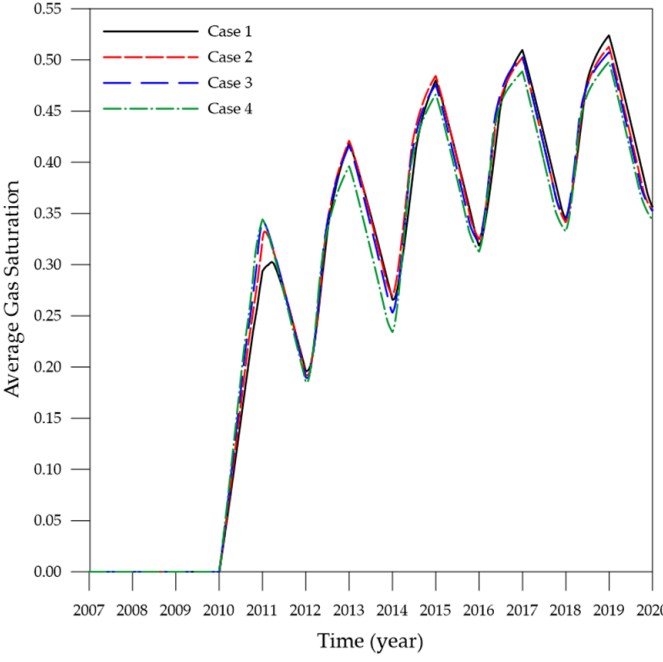

**Figure 7.** Average gas saturation obtained from the four $CO_2$-$CH_4$ WAG cases.

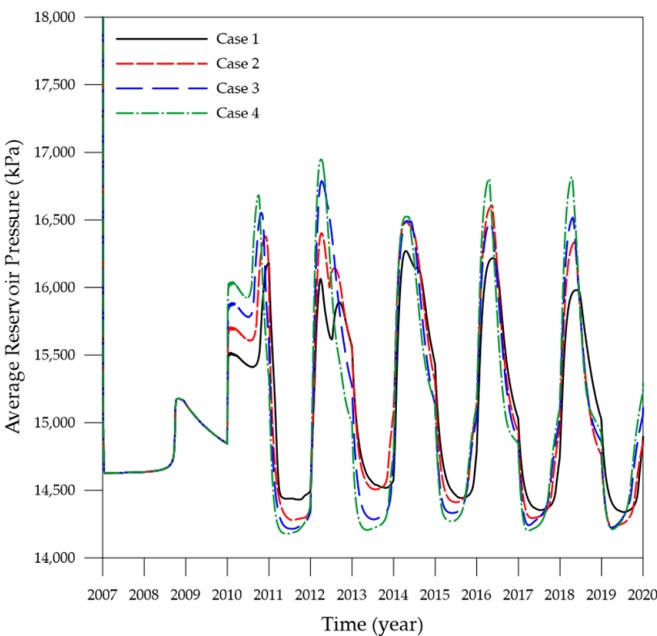

**Figure 8.** Average reservoir pressure obtained from the four $CO_2$-$CH_4$ WAG cases.

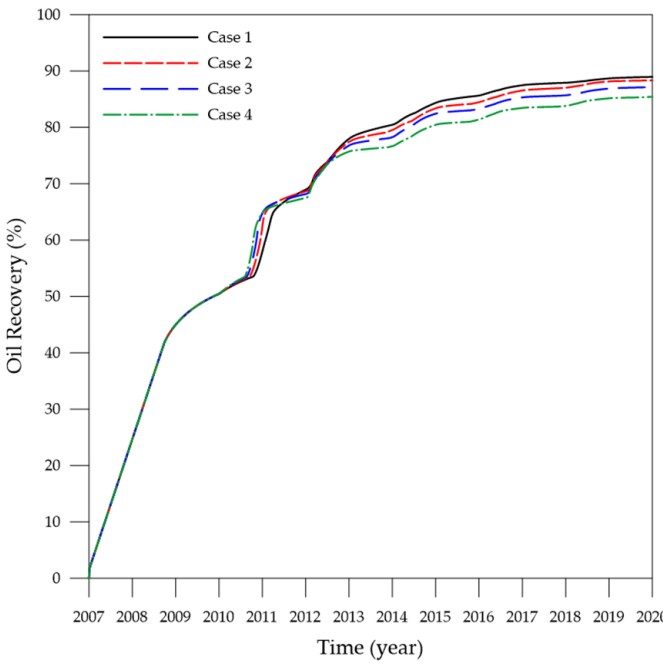

**Figure 9.** Oil recovery factor obtained from the four $CO_2$-$CH_4$ WAG cases.

### 3.3. Effects of $CH_4$ Injection on Carbon Storage Efficiency

As indicated in Figure 7, the addition of $CH_4$ to the $CO_2$ stream reduces the WAG sweep efficiency and gas saturation, thus decreasing the $S_{gm}$ (Equation (4)) to lower the trapped gas saturation, $S_{gr}$. As a result, the residual trapped $CO_2$ by hysteresis (Figure 10) is seen to decrease as the ratio of $CH_4$ to $CO_2$ in the gas stream increases. In addition, the amount of solubility-trapped $CO_2$ decreases as the concentration of $CH_4$ increases, as indicated in Figure 11. However, this decrease is primarily due to the reduced volume of injected $CO_2$ and does not mean that the addition of $CH_4$ lowers the performance of solubility trapping. As mentioned in Section 3.2., the injected total gas rate is set to a constant of 2,265 m³/day, so that the amount of injected $CO_2$ decreases with increasing $CH_4$ addition.

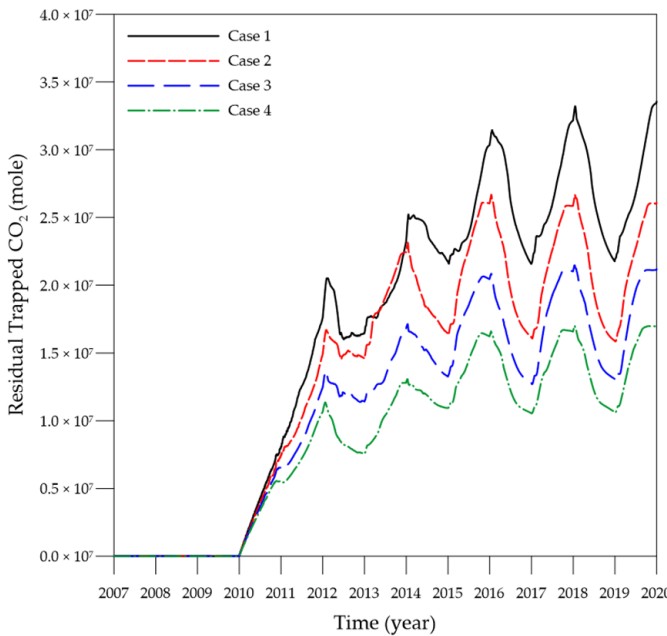

**Figure 10.** Residual $CO_2$ trapped by hysteresis obtained from the four $CO_2$-$CH_4$ WAG cases.

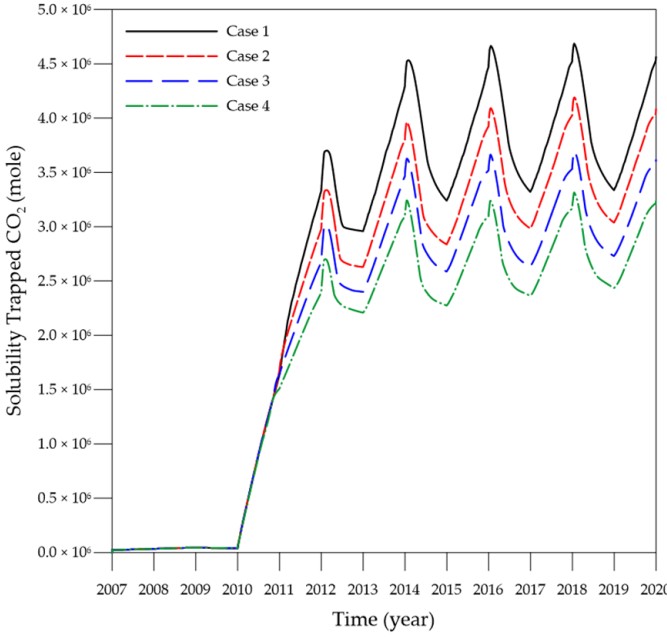

**Figure 11.** Solubility-trapped $CO_2$ obtained from the four $CO_2$-$CH_4$ WAG cases.

The contributions of the residual trapping and solubility trapping mechanisms to geological carbon sequestration is examined in Figure 12 by comparing the mole proportions of injected and remaining $CO_2$. Note that movable $CO_2$ also has the potential to be trapped over time. Thus, compared to Case 1, the residual trapping performances of Cases 2, 3, and 4 are seen to decrease by 13.2%, 20.5%, and 27.4% respectively, due to the lower hysteresis effect and gas saturation. By contrast, the solubility trapping efficiencies remain relatively constant as the amount of injected $CO_2$ is reduced with $CH_4$ addition. This is because the $CO_2$ solubility trapping is affected not only by gas saturation but also by reservoir pressure. Since the added $CH_4$ increases the reservoir pressure, as indicated in Figure 8, this balances out the negative effect of the reduced $CO_2$ injection, thereby resulting in the similar $CO_2$ solubility trapping performance. Meanwhile, the amount of movable $CO_2$ remaining in the reservoir decreases with increasing concentration of $CH_4$ due to the gas breakthrough effect.

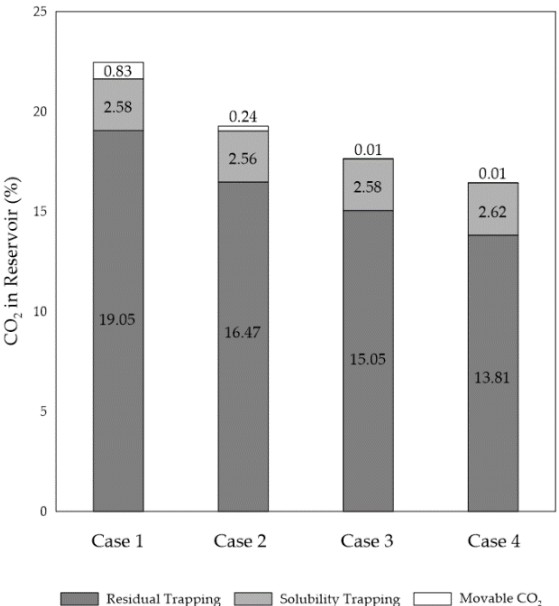

**Figure 12.** The proportion of $CO_2$ remaining in the reservoir relative to the injected volume of $CO_2$ for the four $CO_2$-$CH_4$ WAG cases depending on the trapping mechanisms.

Although the above results elucidate the aspect of $CO_2$ sequestration, it is also necessary to consider the contribution of $CH_4$ storage because the GWP of $CH_4$ is about 28 times that of $CO_2$. The GWP is an index of the relative amount of heat trapped by a unit mass of other GHGs compared to $CO_2$. In the present study, the GWP is transformed into the index per $10^7$ moles of GHG. GWP of $CH_4$ per $10^7$ moles is ten times higher than that of $CO_2$ with consideration for the molecular weight of each gas. In other words, the effect of 1 mole of $CH_4$ sequestration upon the prevention of global warming is the same as that of 10 moles of $CO_2$ sequestration. Thus, the relative amounts of trapped GHGs are presented in Figure 13, where $10^7$ moles of trapped $CO_2$ and $CH_4$ are quantified as 1 and 10, respectively. For Case 1, the net GWP of trapped gases is $3.94 - 2.13 = 1.81$, under the assumption that $CH_4$ is only produced without re-injection into the reservoir. As the $CH_4$ concentration increases, the trapped GWPs for the co-injection cases are increased by 123%, 230%, and 313% respectively, compared with Case 1 due to the $CH_4$ sequestration effect in spite of the reduction in $CH_4$ storage.

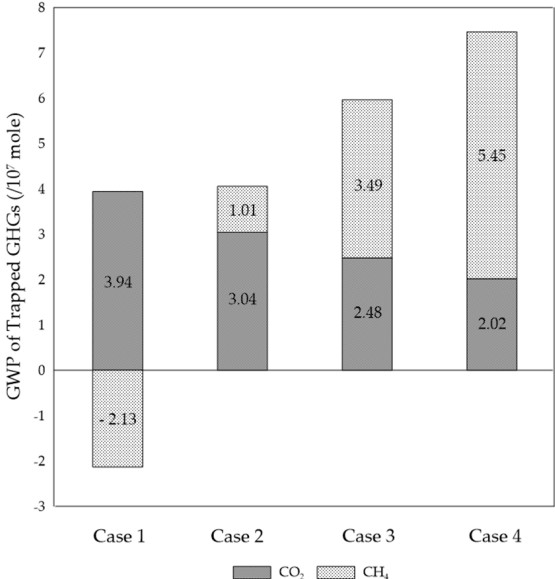

**Figure 13.** The GWP of trapped greenhouse gases obtained from the four $CO_2$-$CH_4$ WAG cases.

## 4. Discussions

This study investigated the effects of $CO_2$-$CH_4$ co-injection on CCS-EOR performance and mechanisms using the compositional 2D reservoir simulation. The 2D homogeneous plane model was designed in order to focus on the miscibility and sweep efficiency without gravity override during WAG simulations. We anticipate that the proposed compositional model can be extended to a three-dimensional (3D) heterogeneous model for real field applications. Adding $CH_4$ into the $CO_2$ stream may affect the vertical sweep efficiency, resulting in the change of the CCS-EOR performance because the density of $CH_4$ is much lighter than that of $CO_2$. Also, reservoirs are heterogeneous in reality. The effects of heterogeneity on the performance are under investigation as future works.

Moreover, it is essential for the technically, economically, and environmentally sustainable development of $CO_2$-EOR fields to analyze both oil recovery and carbon storage efficiencies in an integrated way. For an actual field application, the optimal $CO_2$-$CH_4$ injection design should be developed with consideration for economic factors, such as the volatile oil price, $CO_2$ and $CH_4$ purchase and recycle prices, operating costs, and tax credits for CCS. Ettehadtavakkol et al. [46] focused on the sensitivity of economic parameters with various WAG ratios during CCS-EOR, but no study has been conducted in the $CO_2$-$CH_4$ co-injection system to the best of our knowledge. Since the optimum operating conditions enhancing the performance of CCS-EOR depend on reservoir characteristics, it is necessary to develop a site-specific design based on economic analysis for the field application.

Gas injection changes the stability of asphaltene precipitation and deposition, and formation damages by asphaltene deposition affect CCS-EOR performance. The effects of asphaltene deposition on CCS-EOR in the $CO_2$ WAG process have been investigated in Cho et al. [19], while the effect of $CH_4$ addition has not been studied yet. We clarify that the $CO_2$-$CH_4$ co-injection model considering asphaltene deposition is being developed as our on-going work.

## 5. Conclusions

A compositional numerical simulation was conducted to investigate the effects of $CH_4$ additions upon the coupled $CO_2$-enhanced oil recovery (EOR) and carbon storage process. The oil properties were matched to the reference data from Weyburn W3 via the equation of state (EOS) parameter tuning. The minimum miscibility pressure (MMP) between the $CO_2$/$CH_4$ mixture and the reservoir oil was estimated using the multiple-mixing-cell method. Based on the fluid model, the three-phase hysteresis model and Henry's law were applied for accurate simulation of WAG injection under dynamic conditions. The following conclusions were drawn from this study.

Because adding $CH_4$ to the $CO_2$ stream increased the MMP, the interfacial tension (IFT) between the co-injected gases and the reservoir oil had a higher value than that of $CO_2$ alone. Further, the addition of $CH_4$ resulted in a less pronounced reduction in oil viscosity after multiple contacts with the gas stream, thus indicating the lower displacement efficiency. Since $CH_4$ is less compressible than $CO_2$, the $CO_2$/$CH_4$ mixture occupied more pore volume than $CO_2$ in the first WAG cycle, but the reduction in sweep efficiency due to $CH_4$ addition reversed this tendency in subsequent cycles. Hence, although the $CH_4$ addition resulted in a higher EOR performance in the early stages due to the compressibility effect, the EOR performance was subsequently reduced due to the lower displacement and sweep efficiencies. The diminished sweep efficiency due to $CH_4$ addition led, in turn, to a reduction in the residual trapping performance of $CO_2$. By contrast, the increased reservoir pressure acted to offset the lower sweep efficiency for $CO_2$ solubility trapping, thus maintaining the performance even with increasing amounts of $CH_4$ addition. Taking the global warming potentials (GWPs) of the respective gases into consideration, the overall carbon capture and storage (CCS) effects were improved by 123%, 230%, and 313% by the use of $CH_4$ to $CO_2$ ratios of 0.1, 0.2, and 0.3 respectively, compared to the use of $CO_2$ alone. In conclusion, the developed model demonstrates that $CH_4$ has the combined effect of reducing EOR performance but increasing CCS performance. The above results indicate the significance of the integrated analysis for accurate $CO_2$ sequestration in depleted or depleting hydrocarbon reservoirs under EOR.

**Author Contributions:** Conceptualization, J.C. and K.S.L.; methodology, software, and validation, J.C., K.S.L., and B.M.; formal analysis and investigation, J.C.; writing—original draft preparation, J.C.; writing—review and editing, B.M., G.P., S.K., and H.S.L.; supervision, project administration, and funding acquisition, B.M. All authors have read and agreed to the published version of the manuscript.

**Funding:** This research was supported by the National Research Foundation of Korea (NRF) grants (No. 2018R1A6A1A08025520 and No. 2019R1C1C1002574). Jinhyung Cho was partially supported by the NRF grant (No. 2020R1I1A1A01067015).

**Acknowledgments:** The authors are grateful to the Computer Modelling Group Ltd. (CMG) for technical support.

**Conflicts of Interest:** The authors declare no conflict of interest.

## Nomenclature

| | |
|---|---|
| $\alpha$ | Reduction exponent |
| $f_{i,o}$ | Fugacity of the $i$-th component in the oil phase |
| $f_{i,g}$ | Fugacity of the $i$-th component in the gas phase |
| $f_{i,w}$ | Fugacity of the $i$-th component in the water phase |
| $H_i^S$ | Henry's constant for the $i$-th component at the saturation pressure of water, MPa |
| $k_{rg}^{drain}$ | Relative permeability of the gas during the secondary and following drainage processes |
| $k_{rg}^{input}$ | Input relative permeability of gas |
| $k_{rg}^{imb}$ | Relative permeability of gas imbibition process |
| $n_c$ | Number of components |
| $p$ | Reservoir pressure, MPa |
| $p_c$ | Critical pressure, kPa |
| $p_{H_2O}^s$ | Saturation pressure of water at temperature T, MPa |
| $R$ | Gas constant |
| $S_w^I$ | Water saturation at the start of the drainage process |
| $S_g^{start}$ | Gas saturation at the start of the drainage process (end of previous imbibition) |
| $S_{wi}$ | Initial water saturation |
| $S_{gf}$ | Free gas saturation |
| $S_g^{end}$ | Endpoint gas saturation of imbibition |
| $S_{gc}$ | Critical gas saturation in the input relative permeability table |
| $S_{gm}$ | Maximum gas saturation reached by drainage |
| $S_{g,max}$ | Maximum gas saturation associated with the imbibition |
| $S_{om}^{mod}$ | Modified minimum residual oil saturation |
| $T_c$ | Critical temperature, K |
| $T_{r,H_2O}$ | Reduced temperature of water |
| $\bar{v}_{CO_2}$ | Partial molar volume of $CO_2$ at infinite dilution (cm$^3$·mol$^{-1}$) |

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
