# Peer review of "Compositional Modeling to Analyze the Effect of CH4 on Coupled Carbon Storage and Enhanced Oil Recovery Process"

_applsci, doi:10.3390/app10124272_

Round 1
Reviewer 1 Report
The manuscript: applsci-842531 titled “Effects of CH4 on Displacement and Sweep Efficiencies of a Coupled Carbon Storage and Enhanced Oil Recovery Process”, presents an interesting research study. The manuscript is well written and with proper English. Experimental results and research methods followed were sufficiently described. Figures are well displayed. However, I strongly suggested to distinguish Chapter “Results and Discussions” into two separate Chapters “Results” and “Discussion”. This would significantly contribute in better display the important findings presented in this research paper. In light of the CCS technologies described in this article, you may wish to consider the following paper: “Arvanitis et al. (2020), Potential Sites for Underground Energy and CO2 Storage in Greece: A Geological and Petrological Approach. Energies 13, 2707.” Based on the aforementioned, minor revision is suggested prior to publication in the Journal of Applied Sciences.
Author Response
Please read the attached file.

Reviewer 2 Report
The work herein presented is interesting for publication. I recommend authors the following minor corrections:
The title does not represent the work done. Please find a better title for your work.
Abstract needs to include more data about your results. I think you have a very good results section and you should emphasize it in the abstract.
Please do not include bullet points in the conclusion section. It looks rather an academic work than a paper like that.
Please check some minor grammatical mistakes.
Author Response
Please read the attached file.
